The abundance of large, piscivorous Ferox Trout (Salmo trutta) in Loch Rannoch, Scotland

Thorne Alastair 1
MacDonald Alisdair I. 1
Thorley Joseph L. joe@poissonconsulting.ca 2
1 Freshwater Laboratory, Marine Scotland , Pitlochry , Scotland
2 Poisson Consulting , Nelson , British Columbia , Canada
Audet Céline
Electronic publication date: 2016 Nov 1
Publication date: 2016
Volume: 4
Electronic Location ID: e2646
Received 2016 May 11; Accepted 2016 Oct 3
Copyright: ©2016 Thorne et al.
Copyright year: 2016
Copyright holder: Thorne et al.
License: This is an open access article distributed under the terms of the Creative Commons Attribution License, which permits unrestricted use, distribution, reproduction and adaptation in any medium and for any purpose provided that it is properly attributed. For attribution, the original author(s), title, publication source (PeerJ) and either DOI or URL of the article must be cited.
License URL: https://creativecommons.org/licenses/by/4.0/

Keywords: Survival, Hierarchical, Bayesian, Exploitation, Jolly-Seber, Abundance, Ferox Trout, Brown Trout, Piscivorous

Funding: Poisson Consulting Ltd The study was partially funded by Poisson Consulting Ltd. in the form of a salary for Joseph Thorley. The funders had no role in study design, data collection and analysis, decision to publish, or preparation of the manuscript.

==============================
Background

Ferox Trout are large, long-lived piscivorous Brown Trout (Salmo trutta). Due to their exceptionally large size, Ferox Trout are highly sought after by anglers while their life-history strategy, which includes delayed maturation, multiphasic growth and extended longevity, is of interest to ecological and evolutionary modelers. However, despite their recreational and theoretical importance, little is known about the typical abundance of Ferox Trout.

Methods

To rectify this situation a 16 year angling-based mark-recapture study was conducted on Loch Rannoch, which at 19 km2 is one of the largest lakes in the United Kingdom.

Results

A hierarchical Bayesian Jolly-Seber analysis of the data suggest that if individual differences in catchability are negligible the population of Ferox Trout in Loch Rannoch in 2009 was approximately 71 fish. The results also suggest that a single, often unaccompanied, highly-experienced angler was able to catch roughly 8% of the available fish on an annual basis.

Discussion

It is recommended that anglers adopt a precautionary approach and release all trout with a fork length ≥400 mm caught by trolling in Loch Rannoch. There is an urgent need to assess the status of Ferox Trout in other lakes.

Introduction

Due to its large size and distinctive appearance, the Ferox Trout was originally considered its own species, Salmo ferox (Jardine, 1834); an appellation that was lost when all the forms of Brown Trout were lumped into Salmo trutta. More recently, Duguid, Ferguson & Prodohl (2006) have demonstrated that Ferox Trout in Lochs Melvin (Ireland), Awe and Laggan (Scotland) are reproductively isolated and genetically distinct from their sympatric conspecifics and together form a monophyletic grouping. Based on this evidence, Duguid, Ferguson & Prodohl (2006) argue that the scientific name S. ferox should be resurrected.

Ferox Trout are characterized by their large size and extended longevity. The British rod caught record is 14.4 kg (31 lb 12 oz) and the oldest recorded individual was estimated to be 23 ± 1 years of age based on scale annuli (Campbell, 1979). The consensus view is that Ferox Trout achieve their large size by forgoing spawning until they are big enough to switch to a primarily piscivorous diet at which point they experience an increased growth rate (Campbell, 1971; Campbell, 1979; Went, 1979). The resultant higher survival and fecundity is assumed to compensate for the lost spawning opportunities (Mangel, 1996; Mangel & Abrahams, 2001).

Comparative lake studies (Campbell, 1971; Campbell, 1979) and ecological models (Mangel, 1996; Mangel & Abrahams, 2001) indicate that Ferox Trout require a large (>1 km2) oligotrophic lake and an abundant population of Arctic Charr (Salvelinus alpinus). The ecological models also suggest that under such conditions, Ferox Trout should constitute approximately 5% of the total Brown Trout population (Mangel & Abrahams, 2001). However, there is a lack of robust estimates of the abundance of Ferox Trout or assessments of the potential for angling to impact individual populations. The reasons for this knowledge gap were clearly stated by Duguid, Ferguson & Prodohl (2006, p. 90).

One of the main difficulties in attempting a detailed ferox study is obtaining sufficient specimens. Ferox densities are believed to be low, and their large size and usual distribution deep in the water column makes angling the only practical way to obtain fish. Only a small number of ferox, however, are caught from any lake in a single year even by anglers specializing in ferox capture.

At 19 km2, Loch Rannoch, which is situated in central Scotland, is one of the largest lakes in the United Kingdom. It was chosen for the current study due to its long history of producing Ferox Trout (Campbell, 1971; Campbell, 1979). Whether the Ferox Trout in Loch Rannoch are sufficiently isolated and genetically distinct to be considered a separate species (Duguid, Ferguson & Prodohl, 2006) is unknown. Consequently, Ferox Trout were identified based on their large size and capture method—trolled dead baits and lures (Campbell, 1971; Campbell, 1979; Went, 1979; Grey et al., 2002). As well as Brown and Ferox Trout, Loch Rannoch also contains three ecologically and morphologically distinct forms of Arctic charr (Verspoor et al., 2010).

In 1994, the first author (AT)—a highly-experienced ferox angler—began tagging and releasing all Ferox Trout captured by himself or his boat companion on Loch Rannoch. He continued this practise for 16 years. The paper uses the resultant dataset to estimate the abundance of Ferox Trout in Loch Rannoch. Although the current dataset represents a unique opportunity to better understand the life history of this top-level piscivore, the data are nonetheless sparse. Consequently, they are analyzed using Bayesian methods which provide statistically unbiased estimates irrespective of sample size (Kéry & Schaub, 2011; Royle & Dorazio, 2008).

Methods

Field site

Loch Rannoch, which is located in Highland Perthshire (Latitude: 56.685 Longitude: −4.321), has a length of 15.1 km, width of 1.8 km and a maximum depth of 134 m. It is oligotrophic with a stony shoreline and lies in a catchment dominated by mixed relict deciduous and coniferous woodlands with areas of rough grazing and marginal cultivation. Murray & Pullar (1904) provide a more complete description of its physical characteristics.

Loch Rannoch is part of the Tummel Valley Hydro Electric generation complex and has been a hydroelectric reservoir since 1928, when Rannoch Power Station began to receive water from Loch Ericht. A low barrage at the eastern end of the loch limits the change in water level to a maximum of 2.74 m.

As well as Brown and Ferox Trout, the loch contains at least seven other species of fish: Arctic Charr, Atlantic Salmon (Salmo salar), Pike (Esox lucius), Perch (Perca fluviatilis), Eels (Anguilla anguilla), Three-Spined Sticklebacks (Gasterosteus aculeatus) and Minnows (Phoxinus phoxinus).

Fish capture and tagging

Between 1994 and 2009, AT tagged and released all Ferox Trout captured by himself or his boat companion while angling on Loch Rannoch (Fig. 1). In the absence of any genetic data, a Ferox Trout was deemed to be any member of the Brown trout species complex that was caught by trolling with a fork length ≥400 mm. A fork length of 400 mm was chosen as this is considered to be the upper length threshold for the inferred switch to piscivory (Campbell, 1971; Campbell, 1979). All the fish were caught during the angling season (March 15 to October 6) by licensed anglers using permitted angling methods. The research was conducted within the framework of the UK 1986 Animals Scientific Procedures Act.

Figure 1 Map of Ferox Trout Caught by Angling on Loch Rannoch between 1994 and 2009.

The 69 initial captures are indicated by black circles and the 11 inter-annual recaptures by red triangles. Consecutive recaptures of the same individual are linked by black lines. The coordinates are for UTM Zone 30N (EPSG 32630). Map data from Land Cover of Scotland data, MLURI 1993.

The Ferox Trout were angled by trolling mounted dead baits and lures behind a boat at differing depths and speeds (Greer, 1995). The dead baits (usually Brown Trout or Arctic Charr) were mounted to impart fish-like movement. An echo sounder was used to search the contours of the loch bottom for drop-offs and likely fish holding areas and to ascertain fishing depth. Typically, one entire circuit of the loch’s shoreline excluding the shallow west end, which has an area of 3 km2, was undertaken on each visit.

Hooked fish were played with care and netted directly into a large tank of water before being carefully unhooked. The fish was then transferred into a large fine-mesh keep net (net pen), on the shore closest to the point of capture, where it was allowed to recover before processing. After recovering, the fish was removed from the keep net and placed in a tank containing water and anesthetic (0.05% aqueous solution of 2-pheoxyethanol). When the fish was sufficiently sedated its fork length and wet mass were obtained. The adipose fin was then clipped to aid in the identification of recaptures. In addition, all but one fish (F63) was externally tagged using a Carlin, dart or anchor tag. The tags included the text “REWARD” and a telephone number for reporting. The reward value which was not printed on the tag was five British pounds. The type of tag used depended on which type was available at the time. After tagging, the fish was returned to the keep net to recover and then released from the shore. The entire procedure typically took less than 30 min. The capture location was estimated using a 1:5,000 map.

Five anglers, including AIM, accompanied AT on one or more occasions. On average AT spent 10 days boat angling per year for approximately 10 h per day while fishing three rods although detailed logs of angling effort were not kept. The boat, outboard, rods, reels, line type and dead bait set-up remained constant throughout the study.

Statistical analysis

Fish

Two fish (F53 and F58), which were both recaught once, were excluded from the study because they had a deformed spine and jaw, respectively. After the further exclusion of four intra-annual recaptures, the data set contained information on 80 encounters involving 69 different Ferox Trout (Table 1); seven of which were recaught in at least one subsequent year.

Table 1 Initial captures and subsequent recaptures of Angled Loch Rannoch Ferox Trout by year.

Captures	Year	95	96	97	98	99	00	01	02	03	04	05	06	07	08	09	
7	1994	0	0	0	1	0	0	1	0	0	0	0	0	0	0	0	
6	1995		0	0	0	0	0	0	0	0	0	0	0	0	0	0	
5	1996			0	0	0	0	0	0	0	0	0	0	0	0	0	
2	1997				0	0	0	0	0	0	0	0	0	0	0	0	
5	1998					0	0	0	1	0	0	0	1	0	0	0	
2	1999						0	0	0	0	0	0	0	0	0	0	
12	2000							0	1	1	0	0	0	0	0	0	
6	2001								1	0	1	1	0	0	0	0	
3	2002									0	0	0	0	0	0	0	
4	2003										0	0	0	0	0	0	
2	2004											0	0	0	0	0	
2	2005												0	1	0	1	
1	2006													0	0	0	
1	2007														0	0	
2	2008															0	
9	2009																

Hierarchical Bayesian model

The abundance, annual survival and probability of (re)capture were estimated from the mark-recapture data using a hierarchical Bayesian Jolly-Seber (JS) model (Kéry & Schaub, 2011). The model was the superpopulation implementation of Schwarz & Arnason (1996) in the form of a state-space model with data augmentation (Kéry & Schaub, 2011). Based on preliminary analyses the augmented data set was fixed at 1,000 (genuine and pseudo-) individuals. The zero-inflation of the augmented data set was modeled as an inclusion probability (ψ). Due to the sparsity of data, the annual survival (S) and the probability of (re)capture (p) were assumed to be constant. The only remaining primary parameter was the probability of an individual recruiting to the population at the start of the first year (ρ1). The prior probability distributions for ψ, S, p and ρ1 were all uniform distributions between zero and one. The hierarchical Bayesian JS state-space model made the following assumptions:

1. Every individual in the population had the same constant probability of (re)capture (p).

2. Every individual in the population had the same constant probability of surviving (S).

3. Previously captured individuals were correctly identified.

4. The number of individuals recruiting to the population at the start of each year (B) remained constant.

5. Sampling is instantaneous.

Parameter estimates

The posterior distributions of the parameters were estimated using a Monte Carlo Markov Chain (MCMC) algorithm. To guard against non-convergence of the MCMC process, five chains were run, starting at randomly selected initial values. Each chain was run for at least 105 iterations with the first half of the chains discarded for burn-in followed by further thinning to leave at least 10,000 samples. Convergence was confirmed by ensuring that the Brooks-Gelman–Rubin convergence diagnostic was R ˆ≤1.05 for each of the parameters in the model (Brooks & Gelman, 1998; Kéry & Schaub, 2011). The reported point estimates are the mean and the 95% credible intervals (CRIs) are the 2.5 and 97.5% quantiles (Gelman, 2014).

Software

The analyses were performed using R version 3.3.1 (R Core Team, 2015), JAGS 4.2.0 (Plummer, 2003) and the ranmrdata and ranmr R packages, which were developed specifically for this paper. Article S1 provides instructions on how to download the packages and replicate the analysis.

Results

Fish

The (re)captured fish varied from 400 to 825 mm in length and from 0.62 to 7.41 kg in mass (Fig. 2). Although two large recaptures appeared to senesce (as evidenced by a decline in mass with increasing length), there was no obvious effect of previous capture on body condition (Fig. 2). Tag loss was only recorded for one of the individuals: F21 on its second recapture eight years after it was initially tagged. F21 was identified from photographs of its melanophore constellations (Figs. S1–S3). F13 and F45 were recaught by non-participatory anglers. F45 was released. Both recapture events were excluded from the data, plots and analyses.

Figure 2 Mass-length scatterplot for Ferox Trout Caught by Angling.

The 69 initial captures are indicated by black circles and the 11 inter-annual recaptures by red triangles. Consecutive recaptures of the same individual are linked by black lines.

Parameter estimates

The Bayesian JS mark-recapture model estimated the annual survival (S) to be 0.74 (95% CRI 0.57–0.89) and the annual probability of capture by the primary author or his companion (p) to be 0.08 (95% CRI 0.03–0.16). The inclusion parameter (ψ) was estimated to be 0.42 (95% CRI 0.23–0.76) while the probability of recruiting at the start of the first year (ρ1) was 0.26 (95% CRI 0.13–0.44). The number of individuals recruiting to the population annually (B) was 21 individuals (95% CRI 11–37). The abundance estimate was 111 individuals (95% CRI 39–248) in 1994 and 71 individuals (95% CRI 30 –148) in 2009 (Fig. 3).

Figure 3 Loch Rannoch Ferox Trout abundance estimates by year.

The solid line indicates the point estimates and the dotted lines the 95% credible intervals.

Model adequacy

The Bayesian p-value on the posterior predictive check was 0.31 which indicates that the distribution of the number of encounters (captures and recaptures) each year was consistent with the assumed constant capture efficiency.

Discussion

Abundance

The JS mark-recapture model estimated that the population of Ferox Trout in Loch Rannoch declined from 111 individuals in 1994 to just 71 individuals in 2009. Whether or not the abundance estimates are accurate depends in part on the extent to which the assumption of a constant capture probability is met. The assumption can be violated in two ways: the capture probability can vary among years or it can vary among fish. Variation among years can introduce an artificial trend in the abundance estimates across the course of the study while variation among fish can cause the abundance to be over or underestimated depending on whether any individual differences are fixed (Biro, 2013) or learnt (Askey et al., 2006), respectively. Although angling logs were not kept the posterior predictive check, which compared the number of predicted versus observed encounters, statistically confirmed the relative constancy of p among years. Nevertheless, individual Ferox Trout may still have differed in their vulnerability to capture by angling. As is the case for many mark-recapture studies the reliance on a single capture method and the relatively low number of encounters means it is not possible to determine the presence or form of any individual differences (Biro, 2013).

If the individual differences in catchability are negligible, the abundance estimates are unbiased and by 2009 the Ferox Trout were present at a density of just 0.044 fish.ha−1 when the shallow west end is excluded (Engstrom-Heg, 1986). For comparison, Johnston et al. (2007) estimated that the density of large piscivorous Bull Trout (Salvelinus confluentus) in the 6.5 km Lower Kananaskis Lake, Alberta, was 0.093 fish.ha−1 when being overexploited. In response to a zero-harvest regulation, the density of large Bull Trout in Lower Kannaskis Lake increased to over 2.6 fish.ha−1 in less than a decade. The higher density of Bull Trout in Lower Kananaskis Lake could reflect differences in lake productivity or life-history  strategy.

The annual interval mortality estimate (1 − S) of 0.26 includes handling and tagging by the primary author and his companion as well as natural mortality and fishing mortality by all other anglers on the loch. As all fish recovered well and were only adipose clipped and marked with a single external tag, it is likely that handling and tagging effects were small. Furthermore, despite the offer of a reward only two fish were reported to have been recaught by a member of the public which suggests that the exploitation rate by other anglers on the loch was low. Consequently, if individual differences in catchability are negligible, 26% is probably only a moderate overestimate of the natural mortality rate. For comparison, Johnston et al. (2007) estimated the equilibrium natural mortality rate for adult Bull Trout in Lower Kananaskis Lake to be around 27%.

Management and conservation implications

A concern for any small salmonid population is that the loss of genetic variation results in loss of adaptive potential or inbreeding depression (Wang, Hard & Utter, 2002). Although the levels at which the low genetic variation results in population-level consequences are difficult to predict (Vincenzi et al., 2010), the rate at which genetic variation is being lost can be calculated from the effective population size (Ne) (Wright, 1931; Wright, 1978). Due to their mating systems and life-histories, the Ne of most salmonid populations is considered to be around 25% of the spawning population size (Allendorf et al., 1997; McElhaney et al., 2000). Thus, even if all the adult Ferox Trout in Loch Rannoch spawn in each year then this suggests that if individual differences in catchability are negligible the Ne in 2009 was just eight. The low effective population size is concerning because an Ne ≥ 50 is needed to minimize inbreeding effects and an Ne ≥ 500 is required to retain long-term adaptive potential (Allendorf et al., 1997).

Whether or not the Ferox Trout in Loch Rannoch are at risk of inbreeding depression partly depends on the extent to which they are reproductively isolated from the other Brown Trout in the loch. If they, like the Ferox Trout in Lochs Melvin, Awe and Laggan, are sufficiently isolated and genetically distinct to be considered a separate species (Duguid, Ferguson & Prodohl, 2006) then inbreeding is likely occurring. Alternatively, if the Ferox Trout in Loch Rannoch are simply Brown Trout adopting an alternative life-history strategy, then the effective population size is a function of the total number of Brown Trout spawners and inbreeding is not an issue.

Nonetheless, even if the Ferox Trout in Loch Rannoch are not genetically isolated, a sustained high exploitation rate could result in adaptive change. Mangel and Abrahams’ Mangel & Abrahams (2001) individual-based model predicted that the proportion of the population adopting the ferox life-history strategy is affected by mortality with high size-independent mortality being associated with no or few Ferox Trout. The explanation is straightforward; with increasing mortality the chances of benefiting from delayed maturation diminish. The potentially high catchability suggests that in the absence of catch and release even small amounts of angler effort could produce sufficient fishing mortality to select against the ferox adaptation (Hard et al., 2008).

Given the concerns associated with a potentially high exploitation rate on a long-lived, late-maturing population it is recommended that anglers adopt a conservative approach and release all trout longer than 400 mm caught by trolling in Loch Rannoch. There is an urgent need to assess the status of Ferox Trout in other lakes.

Supplemental Information

Figure S1 Fish 21 on May 19th, 1998. Photograph by Alastair Thorne

Click here for additional data file.

Figure S2 Fish 21 on June 15th, 2002. Photograph by Alastair Thorne

Click here for additional data file.

Figure S3 Fish 21 on July 7th, 2006. Photograph by Alastair Thorne

Click here for additional data file.

Article S1 Instructions on how to replicate the analyses

Click here for additional data file.

We thank the Loch Rannoch Conservation Association for approving the study; C and J Monkton for allowing access to the loch and providing a mooring; PJ Bacon, RA Duguid, R Greer, RL Irvine, IA Malcom and AF Youngson for feedback on earlier drafts and S Vincenzi and two other anonymous reviewers for their helpful comments. We particularly thank the Ferox85 group members who assisted the study.

Additional Information and Declarations

Competing Interests

Author Contributions

Animal Ethics

Data Availability

Alastair Thorne and Alasdair MacDonald are members of Ferox85, an informal organization dedicated to understanding Ferox Trout biology and management. Joseph Thorley is employed by Poisson Consulting Ltd. to provide independent analytic services for a wide variety of government agencies, corporations and conservation organizations on a range of different species and issues.

Alastair Thorne and Alisdair I. MacDonald conceived and designed the experiments, performed the experiments, reviewed drafts of the paper.

Joseph L. Thorley analyzed the data, wrote the paper, prepared figures and/or tables.

The following information was supplied relating to ethical approvals (i.e., approving body and any reference numbers):

All the fish were caught during the angling season (March 15 to October 6) by licensed anglers using permitted angling methods. The research was conducted within the framework of the UK 1986 Animals Scientific Procedures Act.

The following information was supplied regarding data availability:

ranmrdata R data package:

10.5281/zenodo.51110,

https://github.com/Poissonconsulting/ranmrdata;

ranmr R analysis package:

10.5281/zenodo.51274,

https://github.com/Poissonconsulting/ranmr.

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
