# Peer review of "The abundance of large, piscivorous Ferox Trout (Salmo trutta) in Loch Rannoch, Scotland"

_PeerJ, doi:10.7717/peerj.2646_

## Round 0.1 · original submission · Major Revisions

The referees agreed on the interest of this scientific contribution. However, they raised different concerns that should be addressed, especially those that raised reasonable questions about the “speculative” nature of findings. I invite the authors to especially consider all the comments that may help to increase robustness of the paper.

·

Basic reporting

No comments

Experimental design

No comments

Validity of the findings

No comments

Additional comments

I have read with interest the manuscript “The low abundance and high catchability of large, piscivorous Ferox Trout (Salmo trutta) in Loch Rannoch, Scotland”.
The manuscript is interesting, well-written and the authors show mastery of the biological domain and of the statistical tools. I appreciate the detailed, but not overbearing explanation of the statistical methods and the presentation of the code, which also include 2 newly developed R packages. I am familiar with Ferox trout through the work of Marc Mangel’s in the last 90s, early 00s.
I have just a few comments:
- I would rephrase the sentence at lines 37-40. It is quite convoluted and difficult to read
- Line 51-54: Not strictly related to the manuscript, but in this day and age with some funding and now the technology, identifying/developing SNPs for species identification is easy. I strongly recommend the authors to think in this direction
- Figure 1: Can you add a small inlet or outlet with the position of the lake in Scotland?
- Line 63-64: Is the main goal to use Bayesian methods to provide statistically unbiased estimates? Unbiasedness is not the main goal I’d think of.
- Line 80-82. This is related to my previous observation, since identifying ferox trout this way can be misleading and/or circular. From the description here, it sounds like ferox trout is a big brown trout. This is how ferox trout is commonly identified, but I find this way of defining the individual/species quite unsatisfying (even if you add other elements at lines ~214). That’s also why I recommend (not rocket science, I admit) the development of molecular markers.
- Line 155: you use the term “annual probability of capture”, but I do not think it is accurate. The probability of capture is per sampling occasion, if you carried out sampling every two years, you would not have a “bi-annual probability of capture”.
- Figure 3: Have you tried to estimate the parameters of a growth model (von Bertalanffy’s for example)? It would provide a useful reference point (you might have some data points for length/age at length < 400 mm)
- Caveats very well explained in the discussion part
- Figure 4: Credible intervals are very large, but that depends on the sparse data. Please use labels on the x-axis
- Line 212: inbreeding occurs inevitably, but inbreeding depression may not. In some of the salmonid populations I am studying the Fis is closer to 1 than 0.5, but there are no signs of inbreeding depression (purging ecc.)

Reviewer 2 ·

Basic reporting

This study provides a population estimate of the abundance of Ferox Trout, a large long-lived piscivorous Brown Trout (Salmo trutta), in Loch Rannoch, Scotland. In addition, authors try to calculate the catchability of this fish in the same lake. Population abundance was estimated using the data from 16 years of angling in Loch Rannoch with a Jolly-Seber model. Ferox Trout catchability was calculated with hypothetical angler effort and exploitation rate.

Experimental design

I recognize that the 16 years angling database is original and valuable for such a rare form of Brown Trout. However, I cannot recommend this manuscript for publication in its present form, mostly because of two major flaws (see next section).

Validity of the findings

1. It is too bad that no record of angling effort was kept during the duration of the study (lines 105-106; line 168). Authors provide an idea of yearly angling pressure on lines 104-105 but what was the range of the fishing effort (min – max )? Angling effort is a basic variable when dealing with recreational fisheries data. Without angling effort, it is not possible to provide in situ calculation of catchability of Ferox Trout in Loch Rannoch. Consequently, all the section on catchability (lines 182-199) appears to be purely speculative. Hence, I consider that one of the two objectives of the study is not acceptable for publication in its present form. At least, the absence of angling logs does not affect the calculation of population abundance (first objective).
2. The age determination procedure is poorly described. Length-at-age data are very variable (Figure 3), more than what is expected in a single fish population. For example, at 11 year old, fork length of Ferox Trout range from circa 450 to 750 mm. Similarly, at 800 mm, age range from 9 to 19 years. This huge variability may be explained by a problem of age determination. It is well known that scale may underestimate age in long-lived species. On lines 147-149, authors indicate that in some cases, the age determination at recapture correspond to the number of years since the initial capture. In other cases, it does not correspond, apparently due to scale erosion, and authors use the number of years since the initial capture to age the fish. This criteria seems very subjective to me. In the group of 7 fish that were recaptured, how many helped to validate the age determination procedure? How many had the eroded scale? I suggest to include a full description of the age determination procedure in the methods (e.g. area of collection on the fish, handling, mounting, reading and interpretation) and I also suggest to provide more support (data, photos, references) on the age validation using scales in Ferox Trout.

Additional comments

Minor comments:
Line 70. ‘Murray and Pullar (1904)’ instead of ‘Murray and Pullar (Murray and Pullar, 1904)’
Line 115. ‘Schwarz and Arnason (1996)’ instead of ‘Schwarz and Arnason (Schwarz and Arnason, 1996)’
Lines 168-170. This sentence needs more support
Line 182. The methodology of catchability calculation should be in the methods although I do not think that this section should be in the paper.
Line 190 and 193. What is Ferox85 group?
Line263. Salvelinus confluentus shoud be in italic.

Reviewer 3 ·

Basic reporting

no comments

Experimental design

see comments to the author

Validity of the findings

see comments to the author

Additional comments

I enjoyed reading the manuscript “The low abundance and high catchability of large, piscivorous Ferox Trout (Salmo trutta) in Loch Rannoch, Scotland”. The authors present one of the first abundance estimates and catchability estimates of Ferox Trout which is of general interest for the scientific community. The dataset, however, is solely based on angling captures and only one experienced angler was largely involved in data collection. Based on this procedure initial captures and recaptures are likely biased towards the most vulnerable individuals of the population along the favorite fishing track of a single angler. As a result, it seems that fish abundances are underestimated and catchabilities are overestimated. I still think that the manuscript can be published after the following changes:

- Lines 124 – 125: Because individuals likely did not have the same probabilities of capture and survival, model assumptions abet underestimations of Ferox Trout population size. This should be clearly stated already in the methods section.
- Line 181: Densities of Ferox Trout are compared with other large piscivorous salmonids, but conclusions as a result of this comparison are missing. Here it should be again stated that model predictions in the current study likely underestimated Ferox Trout densities.
- Based on the above, statements about catchabilities and effective population sizes are too bold and should be attenuated (this should be done in the abstract and the discussion).
- Line 190: The Ferox85 group was not introduced before.

These changes can easily be done, but from my perspective they are of major importance.

---

## Round 0.2 · accepted · Accept

The reviewers’ comments have been adequately addressed. Thank you for the quality of your revision.

·

Basic reporting

No comments

Experimental design

No comments

Validity of the findings

No comments

Additional comments

The authors responded well to criticism and comment, I have nothing to add.

Reviewer 3 ·

Basic reporting

No comments

Experimental design

No comments

Validity of the findings

No comments

Additional comments

Thanks for revising the mansucript. From my perspective the manuscript is now ready for publication.